# Dienogest May Reduce Estradiol- and Inflammatory Cytokine-Induced Cell Viability and Proliferation and Inhibit the Pathogenesis of Endometriosis: A Cell Culture- and Mouse Model-Based Study

**DOI:** 10.3390/biomedicines10112992

**Published:** 2022-11-21

**Authors:** Hyun Jin Kim, Sung Hoon Kim, Young Sang Oh, Sa Ra Lee, Hee Dong Chae

**Affiliations:** Department of Obstetrics and Gynecology, University of Ulsan College of Medicine, Asan Medical Center, Seoul 05505, Republic of Korea

**Keywords:** dienogest, endometriosis, pathogenesis, stromal cells, cell viability, cell proliferation

## Abstract

Dienogest (DNG) is a therapeutic medication used in endometriosis treatment. Limited data are available regarding its mechanism of action on endometrial cells. Using in vivo and in vitro models, we investigated whether DNG treatment causes significant biological changes in human endometrial stromal cells (ESCs). The markers related to the pathogenesis of endometriosis in ESCs were evaluated using estradiol, tumor necrosis factor alpha (TNF-α), interleukin 1β (IL-1β), and IL-32, administered alone or in combination with DNG. Implanted endometrial tissues were compared between C57BL/6 mice that did or did not receive DNG treatment by using size measurements and immunohistochemistry. A significant decrease in cell viability, protein kinase B (AKT) phosphorylation, and the expression of p21-activated kinase 4 and vascular endothelial growth factor were observed in ESCs treated with estradiol plus DNG. Cell viability, AKT phosphorylation, and proliferating cell nuclear antigen (PCNA) expression also decreased significantly after TNF-α plus DNG treatment. Treatment with IL-1β or IL-32 plus DNG significantly decreased cell viability or PCNA expression, respectively. The size of the implanted endometrial tissue significantly decreased in mice treated with DNG, accompanied by decreased PCNA expression. Thus, DNG may reduce cell viability and proliferation induced by estradiol, TNF-α, IL-1β, and IL-32, and inhibit the endometriosis pathogenesis by decreasing PCNA expression.

## 1. Introduction

Endometriosis is a disorder characterized by the presence of the endometrial stroma and glands outside the uterine cavity [1]. It affects women of reproductive age, causing a myriad of symptoms, including pelvic pain, dysmenorrhea, dyspareunia, and infertility, and thus, significantly impacting patient quality of life [2,3,4]. Within the general population, the prevalence of endometriosis is reported to be 2–10% in women; however, it can reach 50% in infertile women [5,6]. One of the conventional treatments for endometriosis includes surgical excision of the entire lesion followed by postoperative medical treatment, owing to a high recurrence rate. However, surgical excision can be associated with complications and disease recurrence; a considerable number of patients do not respond to treatment or, ultimately, require multiple surgeries [7,8,9,10].

Gonadotropin-releasing hormone (GnRH) analogs represent the most effective medical treatment for endometriosis. These analogs induce hypoestrogenism, provide effective pain relief, and suppress the progression of endometriotic implants [11]. However, GnRH agonists are limited in that they cannot be used for long-term treatment due to their severe hypoestrogenic effects. In fact, a common agreement has been reached indicating that a GnRH agonist treatment that exceeds 6 month needs to be combined with add-back therapy [12]. In contrast, progestins, which are synthetic progesterone, reduce serum estrogen levels by preventing ovulation without causing hypoestrogenism [13]. Progestins also exert progestogenic effects on an estrogen-primed endometrium [14]. Dienogest (DNG), a progestin and 19-nortestosterone derivative, was first approved for the treatment of endometriosis in Europe in 2009. It is highly selective for progesterone receptors (PGRs) and restricts serum estrogen levels by preventing ovulation, which indirectly controls endometriosis. In addition, it promotes apoptosis and reduces the proliferation of endometriotic cells [15,16].

Although many studies have been conducted, the etiologies and mechanism of endometriosis remain unclear. It is widely accepted that endometriosis is an estrogen-dependent chronic inflammatory condition, and several proinflammatory cytokines may play an important role in its pathophysiology. Tumor necrosis factor alpha (TNF-α) and interleukin 1 beta (IL-1β) are proinflammatory factors that have been associated with the progression of endometriosis. Several studies have shown that TNF-α levels are significantly elevated in the peritoneal fluid of women with endometriosis, and it stimulates the proliferation of endometriotic stromal cells (ESCs) [17,18]. Interleukin 32 (IL-32) is a proinflammatory cytokine and an emerging, potential factor in the pathophysiology of endometriosis. In our previous study that used a cell culture and mouse model, high levels of IL-32 were identified in the peritoneal fluid of patients with endometriosis [19]. The mechanism of action of DNG remains unclear. Thus, we decided to use these proinflammatory factors, along with estradiol, to investigate the effect of DNG on human ESCs.

The aim of this study was to investigate the effect of DNG on the biological changes occurring in human endometrial cells through in vitro and in vivo experiments. To ascertain this, we evaluated cell viability, cell invasion, and several markers associated with the pathogenesis of endometriosis, using estradiol and other proinflammatory factors alone or in combination with DNG treatment in ESCs. Furthermore, we compared the size of the implanted endometrial tissues between mice with and without the administration of DNG, and evaluated the expression of cell proliferation factors and hormone receptors via immunohistochemistry.

## 2. Materials and Methods

### 2.1. Sample Collection and Primary Cell Isolation and Culture

Endometrial samples were obtained from women undergoing laparoscopic or transabdominal hysterectomies with a diagnosis of uterine leiomyomas. Women with endometrial abnormalities, adenomyosis or pelvic endometriosis (including ovarian endometriomas), and those who were taking hormonal medication in the preceding three months were excluded from the study. Additionally, endometrial tissues were collected only during the proliferative phases of the menstrual cycle. Written informed consent was obtained from each patient using consent forms, and the protocols were approved by the Institutional Review Board for Human Research of the Asan Medical Center (2014-1165). 

Cell isolation and culture processing were performed using a previously described protocol [20]. After the first passage, ESCs were assayed immunocytochemically using specific cell-surface markers; we have previously shown that the purity of isolated ESCs is above 95% [20,21]. In all the experiments involving ESCs, cells were utilized only after the first passage.

### 2.2. Experimental Setups for Estradiol, TNF-α, IL-1β, IL-32, and DNG Treatment

In each experiment, separate passages of ESCs from different patients were used. After growing the ESCs to a 70% confluence, they were incubated with serum-free, phenol red-free medium (Sigma-Aldrich, St Louise, MO, USA) for 24 h, followed by treatment with estradiol (Sigma-Aldrich, St Louise, MO, USA) alone, or estradiol and DNG (Sigma-Aldrich, St Louise, MO, USA) for 24, 48, and 72 h. Similarly, ESCs were treated with TNF-α (R&D Systems, Minneapolis, MN, USA), IL-1β (R&D Systems, Minneapolis, MN, USA), and IL-32α/γ (R&D Systems, Minneapolis, MN, USA). That is, cells were treated independently with the vehicle (dimethyl sulfoxide (DMSO; Sigma-Aldrich, St Louise, MO, USA), control), estradiol (10^−8^ M), TNF-α (10 ng/mL), IL-1β (10 ng/mL), or IL-32α/γ (25 ng/mL), alone or with DNG (10^−6^ M) treatment.

### 2.3. Cell Viability Assay

The 3-(4,5-dimethylthiazol-2-yl)-2,5 diphenyltetrazolium bromide (MTT) assay (CellTiter 96 Aqueous Cell Proliferation Assay kit; Promega, Madison, WI, USA) was used for cell survival analysis, according to the manufacturer’s instructions. Absorbance was assessed using a microplate reader at a wavelength of 450 nm. Results were expressed as a percentage of the absorbance observed in control (DMSO) cells, normalized to 100%. The cell viability assay was repeated six times with estradiol, five times with TNF-α, eight times with IL-32α, and six times with IL-32γ.

### 2.4. Invasion Assay

A 96-well Transwell plate containing 8 μm pore-size inserts (Corning, Corning, NY, USA) was used for cell invasion analysis. The plate was coated with Cultrex basement membrane extract (Trevigen, Gaithersburg, MD, USA) and the cells were starved in serum-free Dulbecco’s Modified Eagle’s Medium (DMEM) for 18 h. Subsequently, 50 μL of the cell suspension (50,000 cells/well) was seeded in the top chamber, whereas 100 μL of serum-free DMEM with DMSO, DMSO + estradiol (10^−8^ M), or estradiol (10^−8^ M) + DNG (10^−6^ M) was added to the bottom chamber. Plates were incubated in a humidified atmosphere containing 5% CO_2_ at 37 °C for 24 h. The top chamber inserts were washed with a washing buffer to remove non-invading cells, and the inserts were placed on an assay chamber plate to analyze the number of invaded cells. The invaded cells were labeled with 5 μg/mL calcein-AM (Trevigen, Gaithersburg, MD, USA) in the cell dissociation solution at 37 °C for 1 h and then enumerated. This process was repeated six times. The same approach was employed for the assay with TNF-α (10 ng/mL, repeated seven times) and IL-32α/γ (25 ng/mL, repeated six times). Cell invasion was assessed by measuring the absorbance of the samples at 485 nm (excitation) and 520 nm (emission)—while maintaining the same parameters each time—using a Victor X3 multilabel plate reader (PerkinElmer, Waltham, MO, USA).

### 2.5. Enzyme-Linked Immunosorbent Assay (ELISA)

Vascular epithelial growth factor (VEGF) expression was evaluated using an ELISA kit (R&D Systems, Minneapolis, MN, USA), according to the manufacturer’s instructions. Color absorption was measured at 450 nm, and the optical density was corrected at 570 nm and compared with standardized serial dilutions of recombinant human VEGF. The expression of IL-1β/17/32 was also quantitated using the same approach.

### 2.6. Western Blot Analysis

Total proteins from ESCs were homogenized in a cell lysis buffer (Cell Signaling Technology, Danvers, MA, USA) containing a protease inhibitor cocktail (complete mini tablet, Roche, Indianapolis, IN, USA). Protein samples (50 μg) were loaded and electroblotted onto polyvinylidene fluoride membranes (MilliporeSigma, Burlington, MA, USA). The membranes were blocked at room temperature (25 °C) and incubated overnight with primary antibodies raised against proliferating cell nuclear antigen (PCNA), p21-activated kinase 4 (Pak4), protein kinase B (AKT), extracellular-signal-regulated kinase (ERK), and β-actin at 4 °C. After three washes with distilled water, the membranes were incubated with a horseradish peroxidase-conjugated anti-immunoglobulin G secondary antibody (Invitrogen, Carlsbad, CA, USA) and visualized using an enhanced chemiluminescent substrate (Life Technologies, Carlsbad, CA, USA). Densitometric quantification of the protein bands was analyzed using the Multi Gauge Software (Version 2.3, Fujifilm, Tokyo, Japan).

### 2.7. Allotransplantation of Endometrial Tissues in C57BL/6 Mice

In total, 30 five-week-old C57BL/6 mice (SLC Inc., Shizuoka, Japan) were divided into donor (6 mice) and recipient (24 mice) groups, housed in sterile cages with laminar flow-filtered hoods, and provided ad libitum access to rodent laboratory chow (Purina) and reverse osmosis water. All the mice were ovariectomized and administered estrogen pellets, containing 3.2 µg of 17β-estradiol. After seven days of recovery, the mice were anesthetized by inhalation of Alfaxan (100 mg/kg), xylazine (10 mg/kg), and isoflurane (2%). Endometrial tissues were obtained from the uterus of the mice in the donor group. The uteri were divided into the right and left horn, and two identical 3 × 3 mm^2^ pieces of endometrial tissues cut from the same horn were separately implanted onto the peritoneal wall of one mouse in the control (*n* = 12) and one mouse in the DNG-administered groups (*n* = 12). Two weeks after allotransplantation, the mice in the DNG treatment group were administered DNG (1 mg/kg) for two weeks, after which, all 12 pairs of recipient mice were euthanized. The lesions were photographed, removed, and measured before fixation for immunohistochemical staining. The study protocol was approved by the Institutional Animal Care and Use Committee of the Asan Medical Center (2021-12-116).

### 2.8. Immunohistochemistry

Sections from formalin-fixed, paraffin-embedded tissue blocks of endometriotic lesions were cut (3 µm thick) and mounted on glass slides. Immunohistochemistry was performed in accordance with a method described in a previous study [20]. Sections of the lesions were stained using antibodies against PCNA (1:500, Abcam, Cambridge, UK), Ki-67 (1:400, Abcam), Krüppel-like factor 9 (KLF9) (1:400, LSBio, Seattle, WA, USA), estrogen receptor (ER)α (1:400, Abcam, Cambridge, UK), ER β (1:1000, Invitrogen, Carlsbad, CA, USA), and PGR (1:400, Bioss Inc., Woburn, MA, USA). Positive signals were amplified using ultra-VIEW copper, and sections were counterstained with a hematoxylin and blue reagent. Immunoreactivity intensity for each marker was evaluated and analyzed semi-quantitatively, and expressed as an H score, which was used during analysis with the inForm Advanced Image Analysis software (version 2.2, PerkinElmer, Waltham, MO, USA).

### 2.9. Statistical Analysis

All of the data were assessed using the Kolmogorov–Smirnov test to determine if they were normally distributed. Continuous variables were compared using Student’s *t*-test (two groups) or using an analysis of variance and Fisher’s least significant difference post hoc test for pairwise comparisons (three groups) when the data were normally distributed. Variables were compared using the Mann–Whitney *U*-test (two groups) or the Kruskal–Wallis test, followed by the Mann–Whitney *U*-test with a Bonferroni correction (three groups) when the data were not normally distributed. Statistical computations were conducted using the Statistical Program for the Social Sciences software (version 14.0, IBM), with statistical significance defined as a *p* < 0.05.

## 3. Results

### 3.1. Changes in ESCs in Response to Treatment with Estradiol Alone or in Combination with DNG

To evaluate the effect of DNG in comparison to the treatment with estradiol alone, cell viability, AKT phosphorylation, and Pak4 and VEGF expression were analyzed in ESCs (Figure 1a–d). Cell viability increased after 48 and 72 h of treatment with estradiol alone, and decreased significantly after 48 h of treatment with a combination of DNG and estradiol. The p-AKT/AKT ratio and Pak4 expression also significantly decreased after DNG and estradiol treatment for 48 h, when compared to the increased expression of treatment with estradiol alone. Additionally, the ELISA results of estradiol-treated ESCs showed increased VEGF expression, which significantly decreased after DNG and estradiol treatment.

Cell invasion and expression of several ILs were also evaluated to determine the effect of DNG treatment. A similar trend, as that observed for VEGF expression, occurred in cell invasion; however, the difference was not significant. Moreover, the expressions of IL-1β, IL-17, and IL-32 did not differ significantly based on treatment conditions (Figure 1e–g).

### 3.2. Changes in ESCs in Response to Treatment with TNF-α and IL-1β, Alone or in Combination with DNG

Similar to the previous experiments with estradiol, we analyzed ESCs after treatment with TNF-α and IL-1β, the representative cytokines associated with endometriosis, alone and in combination with DNG. ESCs showed increased cell viability when treated with TNF-α alone, which significantly decreased following treatment with DNG plus TNF-α; similar results were obtained for IL-1β (Figure 2a and Figure 3a). Both AKT phosphorylation and PCNA expression showed a significant increase when treated with TNF-α alone, with a significant decrease following treatment with DNG plus TNF-α (Figure 2b,c). Meanwhile, no difference was observed in AKT phosphorylation levels between cells treated with or without IL-1β or with or without DNG (Figure 3b). Although PCNA expression was significantly increased after IL-1β treatment, it was not significantly decreased by the DNG treatment (Figure 3c). Finally, we evaluated the cell invasion treated with TNF-α or IL-1β, and found no significant changes (Figure 2d and Figure 3d).

### 3.3. Changes in ESCs in Response to Treatment with IL-32α/γ Alone or in Combination with DNG

To evaluate the inhibitory effects of DNG in ESCs with the IL-32 treatment, we treated cultured cells with IL-32α or -γ, separately. Cell viability significantly increased after treatment with IL-32α (Figure 4a) or IL-32γ (Figure 4b), and decreased when administered in combination with DNG. Moreover, after 24 h of treatment with IL-32α, PCNA expression exhibited a significant decrease (Figure 4c); however, no such difference was observed following treatment with IL-32γ and DNG (Figure 4d). Cell invasion (Figure 4e,f), ERK (Figure 4g,h) and AKT (Figure 4i,j) phosphorylation, and VEGF expression (Figure 4k,l) were analyzed after treatment with IL-32α/γ alone or in combination with DNG; no significant differences were observed.

### 3.4. Comparison of Endometrial Tissue Implants in Mice with and without DNG Treatment

Endometrial tissues were allotransplanted into 12 pairs of C57BL/6 mice and results were obtained from 9 pairs of mice (total 18 mice). Lesion size was compared between the control and DNG-administered groups. The lesion sizes of the implanted endometrial tissues in mice treated with DNG were significantly smaller than those in the control mice, averaging from 53.70 mm^3^ to 21.46 mm^3^ (Figure 5a). In addition, we evaluated the inhibitory effect of DNG as a percentage of direct pairs that shared the same horn of the uterus. The inhibitory effect of DNG showed a more significant decrease in lesion size (61.42% when regarding the control as 100%) and was statistically significant (Figure 5b).

### 3.5. Expression of Several Markers in Endometrial Tissues Implanted in Mice with and without DNG Treatment

Using the mouse model experiment, we performed immunohistochemical staining of the implanted endometrial tissues using several endometriosis-related factors. The expression level of PCNA, a cell proliferative factor, was significantly decreased in glandular and whole lesions of mice treated with DNG compared with that of control mice (Figure 6a). A similar decreasing trend was observed for nuclear protein Ki-67 (Figure 6b) and Krüppel-like factor 9 (KLF9) (Figure 6c) expression in glandular and whole lesions of mice treated with DNG; however, the differences were not statistically significant. Moreover, the expression of estrogen receptor (ER)α/β and its ratio (Figure 6d,e), as well as the expression of PGR (Figure 6f), increased in both groups; however, the changes in ER expression and ratio, as well as in PGR expression, were not significant.

## 4. Discussion

The present study showed that under in vitro conditions, DNG induced a decrease in cell viability and proliferation with increasing levels of estradiol, TNF-α, IL-1β, and IL-32. Furthermore, our mouse model demonstrated that the size of the implanted endometrial tissues was smaller in mice treated with DNG than in mice without the DNG treatment, and there was a decreased expression of PCNA without any significant changes in the expression of hormone receptors. These in vitro and in vivo data showed the effects of DNG on human ESCs and its anti-inflammatory characteristics, as well as the changes in lesions occurring in endometriosis.

The etiology of endometriosis remains unclear, and many researchers have attempted to determine the mechanism underlying endometriosis development. Diverse cytokines, chemokines, and pathways have been shown to be involved in the development of endometriosis. However, it is assumed that endometriosis could potentially be an estrogen-dependent disease, related to chronic inflammation [22,23,24,25]. When ESCs were treated with estradiol, we observed an increase in viability, AKT phosphorylation, and Pak4 and VEGF expression. Upon treatment with both estradiol and DNG, cell viability decreased, as did AKT phosphorylation and Pak4 and VEGF expression, demonstrating the inhibitory effect of DNG in human ESCs. However, the same was not observed for the expression of interleukins (IL-1β, 17, and 32), the levels of which are elevated in women with endometriosis or ectopic lesions [19,26,27]. In addition, no unified results were observed based on the time of treatment; however, the most significant results were obtained after 24 or 48 h. This might be due to a difference in the experimental setup, wherein we subjected human ESCs to treatment with both estradiol and DNG, in contrast to using direct ectopic lesions or the peritoneal environment. Additionally, with time, human ESCs can become refractory to treatment with estradiol or the inflammatory cytokines used in this study.

Additionally, we elected to use endometrial stromal cells in the current study rather than endometrial epithelial cells. Although both cell types are separate, yet still interact with each other, stromal cells are suspected of having a more prominent role in endometrial growth and differentiation [28,29]. Moreover, given that a significant pathogenic factor of endometriosis is progesterone-resistant dysregulation of decidualization, which is the process of stromal cell transformation, employing ESCs in our study facilitated the analysis of common characteristics of endometriosis [30,31].

The inflammatory cytokines, TNF-α and IL-1β, play crucial roles in triggering the inflammatory pathway and are upregulated in the peritoneal fluid of women with endometriosis. These cytokines promote ESC proliferation and induce the upregulation of nerve growth factor, which may represent the primary mechanism by which DNG relieves endometriosis-associated pain. Additionally, TNF-α and IL-1β affect nuclear signaling, which may contribute to the pathogenesis of endometriosis [17,32,33,34]. The current study results indicate that cell viability, AKT phosphorylation, and PCNA expression were increased following the treatment of ESCs with TNF-α, all of which were decreased after the addition of DNG. Similarly, a significant increase was observed in cell viability after IL-1β treatment, which was decreased following DNG addition.

IL-32 is also a proinflammatory cytokine that serves as a potent inducer of other cytokines, including TNF-α, as well as a controller of immune function, host defense, and cell death [35,36,37,38]. IL-32 is involved in several chronic inflammatory diseases, including chronic rhinosinusitis, ankylosing spondylitis, and inflammatory bowel diseases [39,40,41]. Moreover, IL-32 levels are reportedly elevated in the serum and peritoneal fluid of patients with endometriosis [42]. In fact, we previously reported a significant increase in ESC viability following treatment with IL-32α and IL-32γ [19]. In the current study, we investigated the effect of IL-32α and IL-32γ under DNG treatment to determine the factors or pathways associated with its inhibitory effects. We observed a significant decrease in cell viability and PCNA expression, particularly with IL-32α, whereas other factors, such as cell invasion, AKT/ERK phosphorylation, and VEGF expression, did not exhibit significant changes. These results suggest that IL-32 may have an important role in the pathogenesis and/or pathophysiology of endometriosis; however, it is not closely related to the DNG mechanism of action.

The present study used a mouse model to confirm the lesion-reducing effect of DNG. As mentioned, DNG is now the first-line regimen for endometriosis treatment, i.e., to inhibit the growth of proliferative lesions and to manage the associated symptoms. A 2 mg-per-day dose of DNG is the standard that demonstrates significant efficacy with respect to lesion and pain reduction and is considered superior to the placebo; however, it is equivalent to the GnRH analog [43,44,45,46,47]. We observed a gross reduction in lesion size (Figure 6) that was significant when compared between paired mice that shared the same uterine horn of the donor mouse. The dose of DNG administered to the mice was considerably higher than the daily dose of DNG recommended for use in humans; however, a higher dose was used with the aim of more precisely confirming the lesion reduction effect.

DNG is a synthetic progestin and a selective PGR agonist. Pharmacodynamically, it has moderate affinity for PGR and activates neither ERα nor -β [48,49]. Although the precise pathogenesis of endometriosis remains unclear, hormonal influences and genetic/epigenetic factors are expected to justify the impairment of cellular mechanisms. In fact, estrogen-mediated alterations may play a role in the pathophysiology of endometriosis, especially alterations in ER activity and an increased ERβ/ERα ratio in endometriotic cells, thereby triggering proinflammatory factors and cell proliferation [50,51]. Hence, we performed immunohistochemistry to assess ERα and Erβ expression in the allotransplanted endometrial tissue of mice treated with DNG; however, there was no reverse effect of DNG on the ERβ/ERα ratio. Meanwhile, Hayashi et al. [52] reported a decrease in the ERβ/ERα ratio with DNG treatment; however, they used an in vitro human endometrial cell model, whereas we applied a mouse model. In addition, the increased expression of ERβ in the endometriotic tissue downregulates PGR abundances [51,53]. However, in the current study, immunohistochemistry did not reveal any significant difference in PGR expression between lesions with and without DNG treatment, even though the H-SCORE was higher than that of the control. There are two isoforms of PGR, A and B, which bind distinct but overlapping genomic sites and interact to differentially modulate estrogen signaling [54]. Hayashi et al. [52] reported an increased ratio of PGR B/A by using a DNG treatment for human ovarian endometriosis. Although we were unable to differentiate between the two isoforms of PGR, our immunohistochemistry data also showed an increased tendency in PGR expression in those treated with DNG. Hence, taken together, these findings suggest that DNG treatment may overcome progestin resistance in endometriosis.

We also analyzed the expression of PCNA, Ki-67, and KLF9 using immunohistochemistry; however, only the expression of PCNA was significantly impacted by DNG treatment. Nevertheless, the transcription factor, KLF9, has emerged as a potent player in reproductive dysfunctions associated with aberrant ER and/or PGR signaling, and several studies have investigated its relationship with endometriosis [55,56,57]. Heard et al. [58] showed that endometrial KLF9 deficiency promotes the establishment of endometriotic lesions, whereas the whole, and glandular, H-SCOREs of KLF9 were decreased in mice treated with DNG, with no significance. In this study, we evaluated the factors that are known to be associated with the pathogenesis of endometriosis. Future studies should focus on identifying anti-inflammatory targets to further elucidate the mechanism underlying the development of endometriosis and its treatment with DNG.

This study has successfully gathered extensive data—from cell culture- to mouse model-based research—and confirmed the association between several endometriosis-associated factors and the therapeutic mechanism of DNG. However, there are several limitations that warrant discussion. First, we evaluated several factors associated with inflammation and endometriosis; however, we were unable to define a specific factor closely related to the pathophysiology of endometriosis or the DNG mechanism of action. This limitation can be justified by the fact that inflammation is not a linear process, but rather a complex network, making it difficult to identify a single most critical factor. Second, cell culture and immunohistochemistry findings from the mouse model provided limited data, with only PCNA expression exhibiting significant changes in the whole and glandular regions. This may be due to the different ecologies between humans and mice, as well as the high dose of DNG used. In addition, the results may have been influenced by unknown differences between the subjects, such as disease activity, type of the lesions, menstrual phase, and genetic variation. Lastly, this study evaluated the effects of DNG alone and did not consider other types of progestins, such as medroxyprogesterone acetate or levonorgestrel. Importantly, further studies are needed to investigate the inhibitory effect of DNG based on distinguishable characteristics, in addition to experiments using other progestins.

## 5. Conclusions

The findings of this study suggest that DNG may reduce cell viability and proliferation in response to treatment with estrogen, TNF-α, IL-1β, or IL-32, and inhibit the pathogenesis of endometriosis by decreasing PCNA expression.

## Figures and Tables

**Figure 1 biomedicines-10-02992-f001:**
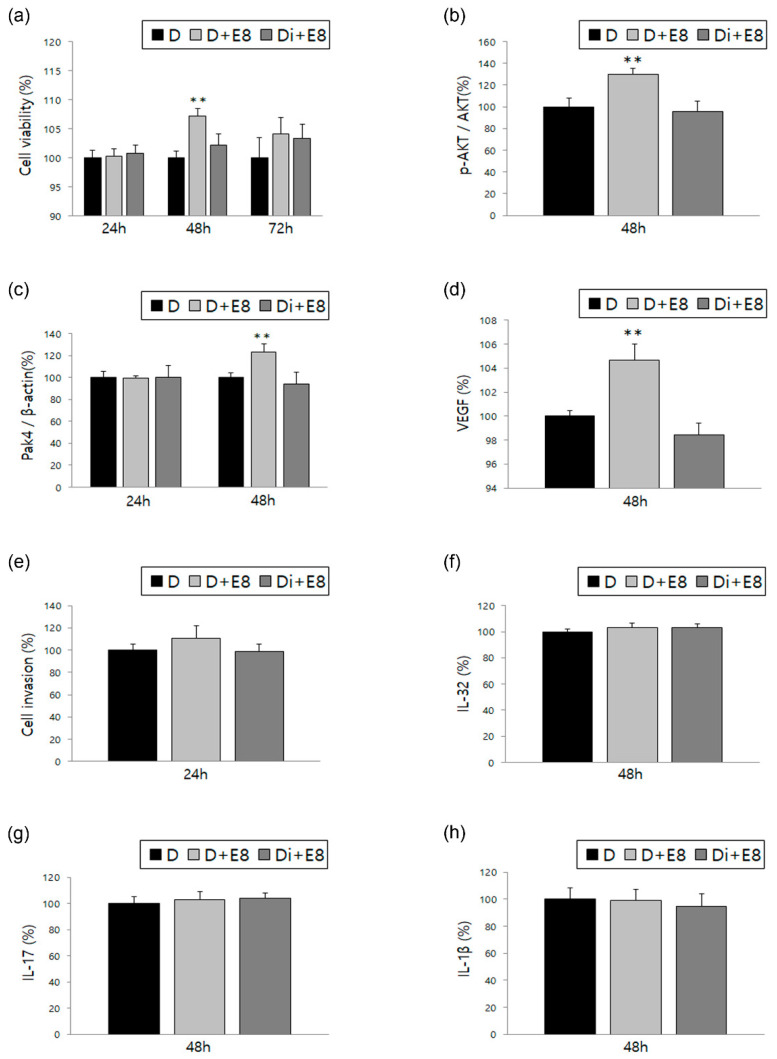
Effect of dienogest (DNG) and estradiol on human endometrial stromal cells (ESCs). Time-dependent variation evaluated in the three groups for the following factors: (**a**) cell viability, (**b**) AKT phosphorylation, (**c**) Pak4 expression, (**d**) VEGF expression, (**e**) cell invasion, (**f**) IL-32 expression, (**g**) IL-17 expression, and (**h**) IL-1β. Error bars show the mean ± standard error of mean (SEM). ** *p* < 0.05 versus control as well as treatment with both DNG and estradiol in ESCs. D = ESCs in dimethyl sulfoxide solution (DMSO; control); D + E8 = treated with estradiol 10^−8^ M; Di + E8 = treated with DNG 10^−6^ M and estradiol 10^−8^ M. Data are expressed as a percentage, wherein cells treated with the vehicle are normalized to 100%.

**Figure 2 biomedicines-10-02992-f002:**
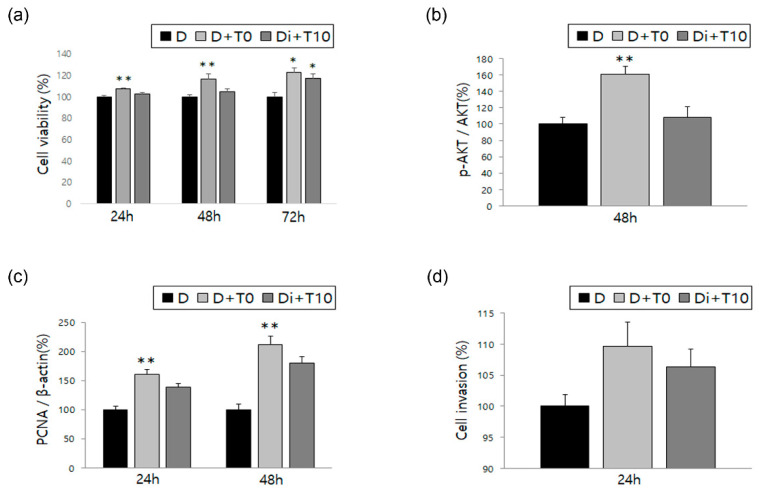
Effect of DNG with tumor necrosis factor alpha (TNF-α) on human ESCs. Time-dependent variation evaluated in the three groups for the following factors: (**a**) cell viability, (**b**) AKT phosphorylation, (**c**) expression of PCNA, and (**d**) cell invasion. Error bars show the mean ± SEM. * *p* < 0.05 versus control, ** *p* < 0.05 versus control as well as treatment with both DNG and TNF-α in ESCs. D = ESCs in DMSO solution (control); D + T10 = treated with TNF-α 10 ng/mL; Di + T10 = treated with DNG 10^−6^ M and TNF-α 10 ng/mL. Data are expressed as a percentage, wherein cells treated with the vehicle are normalized to 100%.

**Figure 3 biomedicines-10-02992-f003:**
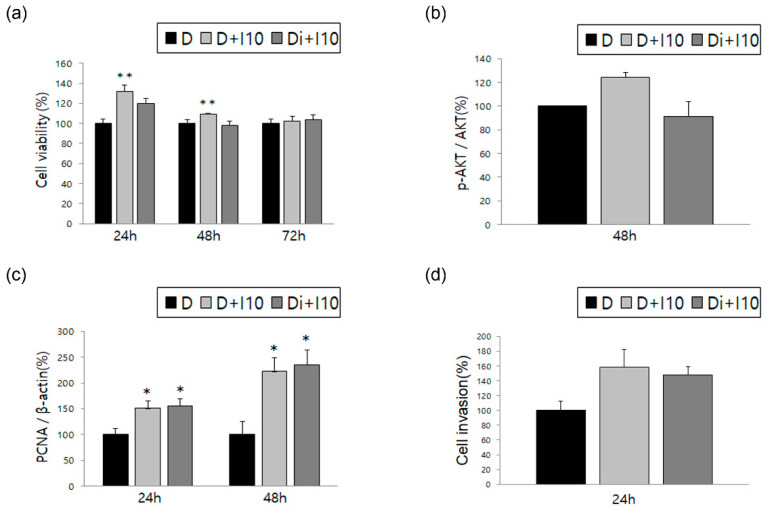
Effect of DNG with interleukin (IL)-1β on human ESCs. The graphs show time-dependent variation evaluated in the three groups for the following factors: (**a**) cell viability, (**b**) AKT phosphorylation, (**c**) PCNA expression, and (**d**) cell invasion. Error bars show the mean ± SEM. * *p* < 0.05 versus control, ** *p* < 0.05 versus control as well as treatment with both DNG and IL-1β in ESCs. D = ESCs in DMSO solution (control); D + I10 = treated with IL-1β 10 ng/mL; Di + I10 = treated with DNG 10^−6^ M and IL-1β 10 ng/mL. Data are expressed as a percentage, wherein cells treated with the vehicle are normalized to 100%.

**Figure 4 biomedicines-10-02992-f004:**
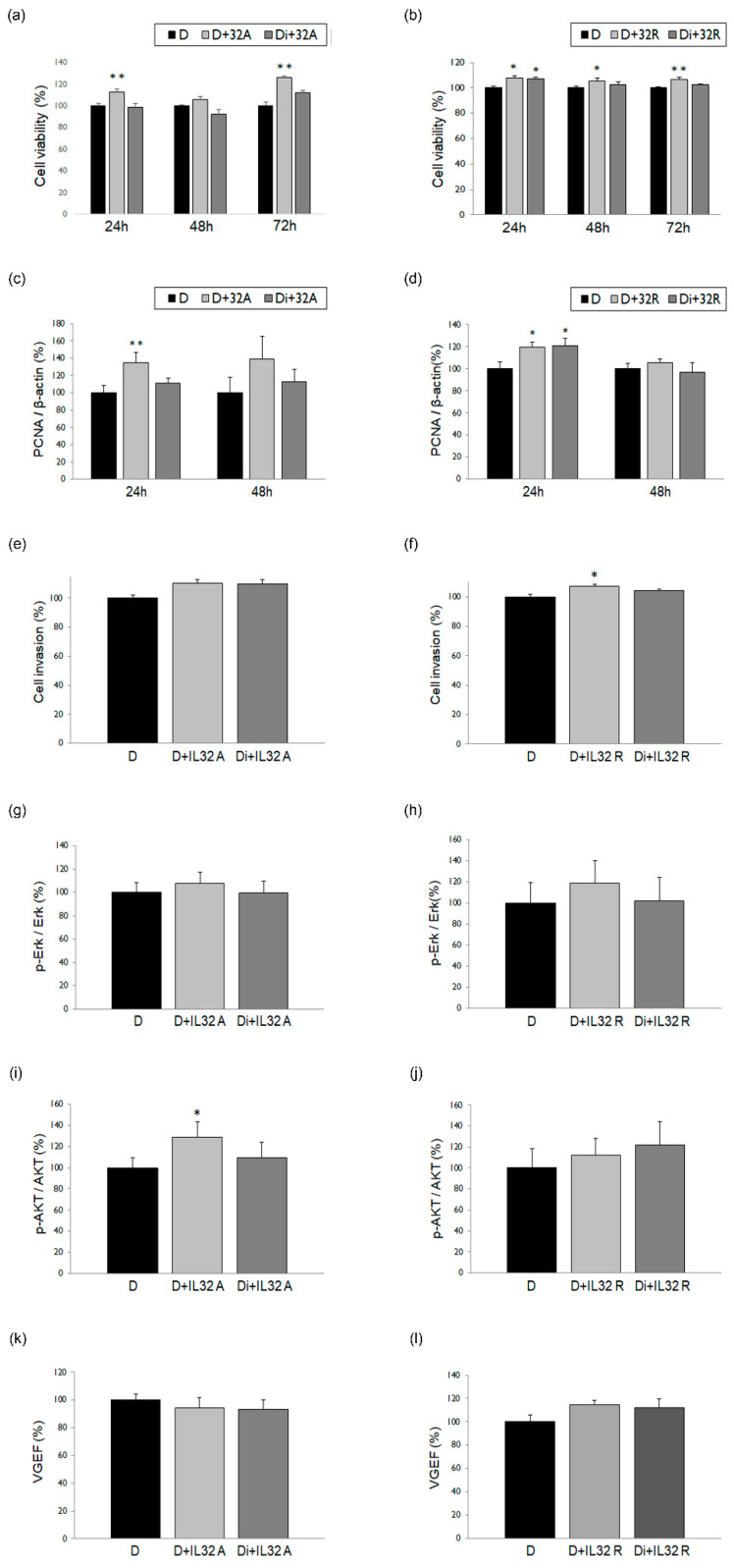
Effect of DNG treatment with IL-32α or IL-32γ on human ESCs. Cell viability, expression of PCNA, cell invasion, ERK and AKT phosphorylation, and VEGF expression following IL-32α (**a**,**c**,**e**,**g**,**i**,**k**) or IL-32γ (**b**,**d**,**f**,**h**,**j**,**l**) treatment alone or in combination with DNG in ESCs. Error bars show the mean ± SEM. * *p* < 0.05 versus control, ** *p* < 0.05 versus control as well as with both DNG and IL-32 α/γ treatments in ESCs. D = ESCs in DMSO solution (control); D + 32A/D + 32R = treated with IL-32α/IL-32γ (25 ng/mL); Di + 32A/Di + 32R = treated with DNG 10^−6^ M and IL-32α/IL-32γ (25 ng/mL). Data are expressed as a percentage, wherein cells treated with the vehicle are normalized to 100%.

**Figure 5 biomedicines-10-02992-f005:**
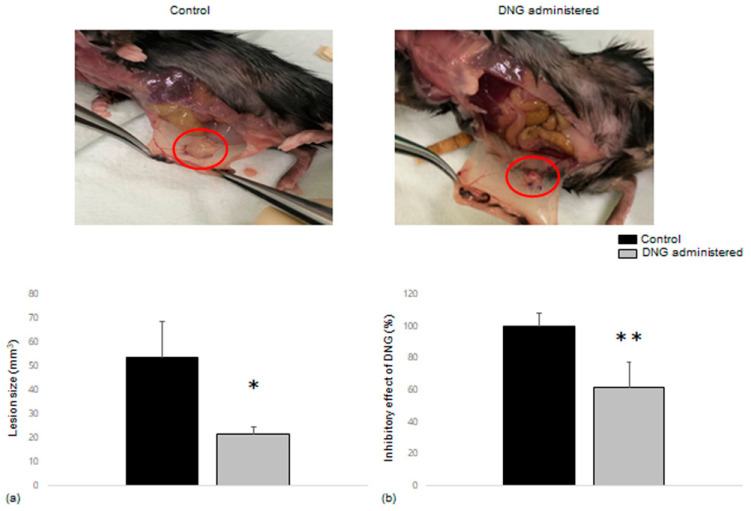
Effect of DNG on ectopic endometrial tissue growth in mice. (**a**) Image and graph showing ectopic endometrial tissues in control mice and DNG-treated mice. (**b**) The size change of the implanted tissues and paired comparison allotransplanted from the same uterus of a mouse, showing the inhibitory effect of DNG. * *p* = 0.049 vs. control. ** *p* = 0.019 vs. control.

**Figure 6 biomedicines-10-02992-f006:**
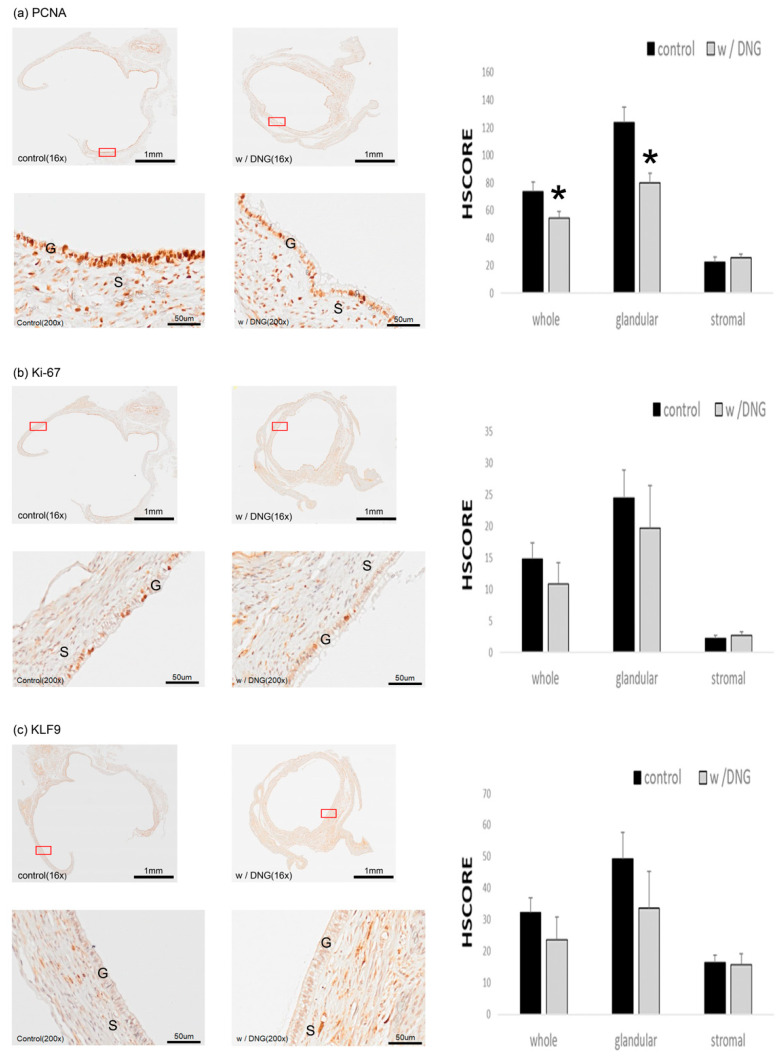
Effects of DNG treatment in mouse model. Representative immunostaining micrographs (×16 and ×200) and HSCOREs for (**a**) proliferating cell nuclear antigen (PCNA), (**b**) Ki-67, (**c**) Krüppel-like factor 9 (KLF9), (**d**) estrogen receptor (ER)α, (**e**) ERβ and ERβ/ERα ratio, and (**f**) progesterone receptor (PGR) in allotransplanted endometrial tissues of mice treated with or without DNG. Values are expressed as the mean ± SEM. * *p* < 0.05 vs. control. G—glandular portion, S—stromal portion.

## Data Availability

Not applicable.

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
