# Peer review of "Dienogest May Reduce Estradiol- and Inflammatory Cytokine-Induced Cell Viability and Proliferation and Inhibit the Pathogenesis of Endometriosis: A Cell Culture- and Mouse Model-Based Study"

_biomedicines, 2022, doi:10.3390/biomedicines10112992_

Round 1

Reviewer 1 Report

The study by Kim et al. investigates the therapeutic mechanisms of Dienogest for the treatment of endometriosis using in vitro and in vivo models. The study is well designed and findings are clearly presented. This research will be of interest to many in the field and I have only minor comments.

1.       The authors have chosen to focus on endometrial stromal cells and not endometrial epithelium. Is there a reason why epithelium was excluded from the in vitro work? A short rational for this would be useful in the introduction.

2.       Materials and methods - The cycle phase of endometrial tissue donors should be stated

3.       Line 212 ‘there were no significant changes in AKT phosphorylation and PCNA expression based on IL-1b treatment’. Figure 3c shows a significant increase in PCNA expression upon IL-1b treatment

4.       Line 270-271 – the authors state that ‘expression level of PCNA was significantly higher in glandular and whole lesions of mice treated with DNG’, whilst Figure 6a shows the opposite effect.  

5.       Figure 6 – IHC image resolution must be improved to better appreciate the staining patterns. Glandular structures are not visible in the images – examples of glandular staining should also be included. Images also require a scalebar.

6.       One would expect a progestin to cause downregulation of PGR, which is not observed in the excised lesions from the mouse model. In fact, PGR expression is higher with DNG treatment (although not significantly so). Can the authors comment on this?

7.       Line 316 – ‘The data showed a significant decrease in cell viability after IL-1b treatment’. This is the opposite of what is shown in Figure 3a.

Reviewer 2 Report

The paper is very interesting. It falls within the scope of this jorunal.

Here my concerns:

Useful for your introduction section: doi: 10.1126/scitranslmed.abd6469; doi: 10.1080/17425255.2020.1789591.

methods and results section are well written, also the figures are very explanatory

What do you think about the possible immunosuppression of T-Reg CD4+ Cells in patients with endometriosis? And dienogest could act on them?

Are there further similar cases of female benign chronic disease in literature?
